# Research Professors' Self-Assessment of Competencies

**Gabriela Torres Delgado *** and Neil Hernández-Gress

Director of Human Development for Research, Research Direction, Tecnologico de Monterrey,
64849 Monterrey, Mexico; ngress@tec.mx
***** Correspondence: gtorresd@tec.mx; Tel.: +52-1-81-2202-9944

**Abstract:** Research professors develop scientific products that impact and benefit society, but their competencies in doing so are rarely evaluated. Therefore, by employing a mixed two-stage sequential design, this study developed a self-assessment model of research professors' competencies with four domains, seven competencies, and 30 competency elements. Next, we conducted descriptive statistical analysis of those elements. In the first year, 320 respondents rated themselves on four levels: initial, basic, autonomous, and consolidated. In the assessment model's second year, we compared 30 respondents' results with those of their initial self-assessment. The main developmental challenge was Originality and Innovation, which remained at the initial level. Both Training of Researchers and Transformation of Society were at the basic level, and Digital Competency was at the autonomous level. Both Teaching Competence and Ethics and Citizenship attained the consolidated level. This information helps establish priorities for accelerating researchers' training and the quality of their research.

**Keywords:** research professors; assessment; model of competencies; transformation of universities; educational innovation; higher education

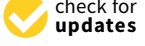



## 1. Introduction

Transformation of others' professional behaviors implies complex work in which varied competencies are collected, analyzed, and validated by international organizations [1–3]. Previously, efforts have been primarily devoted to the evaluation of student competencies, but professors' competencies have recently become very important [4–7] Thus, various models of researcher development have been produced [4,8] to support evaluation of professors [4,5,9–11].

However, instruments for assessing scientific research-focused competencies are not available [12], and, in México, we found few development models for research professors [13] and few studies of the competencies of undergraduate researchers' professors [14,15] despite their great importance. Studies from other countries on researcher development models were found, for instance, [4,9,10,16,17], who present complex models explaining professors' defined behaviors and commenting that doing so is not just simple but also very relevant. This study attempts to evaluate research-professor competencies that demonstrate the following: leadership in a discipline, production of quality research, innovation, teaching, ethics, citizenship, use of technology, and linkage, funding, and training of other researchers—all focused on solving society's current and future problems.

### Framework

The Educational Model of the educational institution where this research was conducted integrates development of researcher-professors as part of its 2030 strategy. In Quacquarelli Symonds (QS) World University Rankings 2020 [18], the institution had advanced 20 positions over the previous year, standing out in México as one of the best private universities. In its new Latin American ranking and for the third consecutive year, Times Higher Education (THE) places the institution as the nation's best university and

the region's fifth best. According to the QS ranking, global employers rank the institution as one of the universities that produces the best graduates for the labor market. Furthermore, the institution climbed 20 places in the category of academic reputation and ranked excellent in Latin America in the category of international professors.

The institution has more than 90,000 students at the high school, undergraduate and graduate levels at 26 campuses throughout Mexico. In the January–May semester 2020, there were 6630 professors, 994 full-time professors, and 637 México's National Research System-research professors. In the same semester, 15,453 students were conducting research. In the August–December semester 2020, there were 508 doctoral students, 2117 master's students, and 42 research groups. From 2014 to 2018, 4397 scientific studies were published, there were 18,372 citations, 107 patent applications, and 102 patents granted. With other top world universities, we conduct collaborations and alliances that promote joint projects.

Professors are considered trainers, advisors and mentors to all those who want to learn. They teach by example, bringing real-world experiences and challenges into the process. A research professor is a professor who decides to direct his or her career to re-search. To be part of the Research Professor Model, it is necessary to be a full professor. A full-time professor is devoted the 100% of his time is for teaching and research activities at the University.

Under these premises, a competence is conceived as the ability to deal successfully with present and future situations, both structured and uncertain, from a basis of complex individual know-how. This know-how results from conscious integration, mobilization, and adaptation of skills, abilities, attitudes, and of values of cognitive, affective, and declarative knowledge of a psychomotor or social nature [19]. A model specifies the competency's (scenario model's) performance levels and its changes through training and time (development model) [4]. With these givens listed above, the institution's faculty should have or develop the following main characteristics:

- The ability to work interdisciplinary: Society's challenges are complex, requiring approaches from multiple perspectives and various fields of knowledge.
- The capacity to develop students' competencies and skills, providing them with the knowledge and tools to approach complex problems with social awareness.
- The ability to generate, apply, and transfer knowledge, thus positively affecting students' integral formation and providing original, innovative solutions to society's problems within the economic, political, and social context.
- The global vision and qualifications to generate opportunities for both students and professors' growth and development through observation of diverse practices from various cultures in research, education, and management.
- Commitment to social responsibility, integrating into research and instruction development of solidarity, respect for human rights, and human dignity within ethical principles.

Considering these characteristics, we established a model for faculty development that focuses on research and outreach, academic leadership, influence and impact on society, and personal and professional excellence. The Institute has focused on promoting research and technological development to encourage the knowledge-based economy; generating business incubation models; collaborating in the improvement of public administration and public policies; and creating innovative models and systems for sustainable.

In the following Figure 1, we would like to present the Model for Research Professors' Competencies development at the institution. As it is integrated by four domains and six competencies, which, as we can review in detail, are shown in Table 1, and which were very carefully considered, and which we will explain in the methodological part.

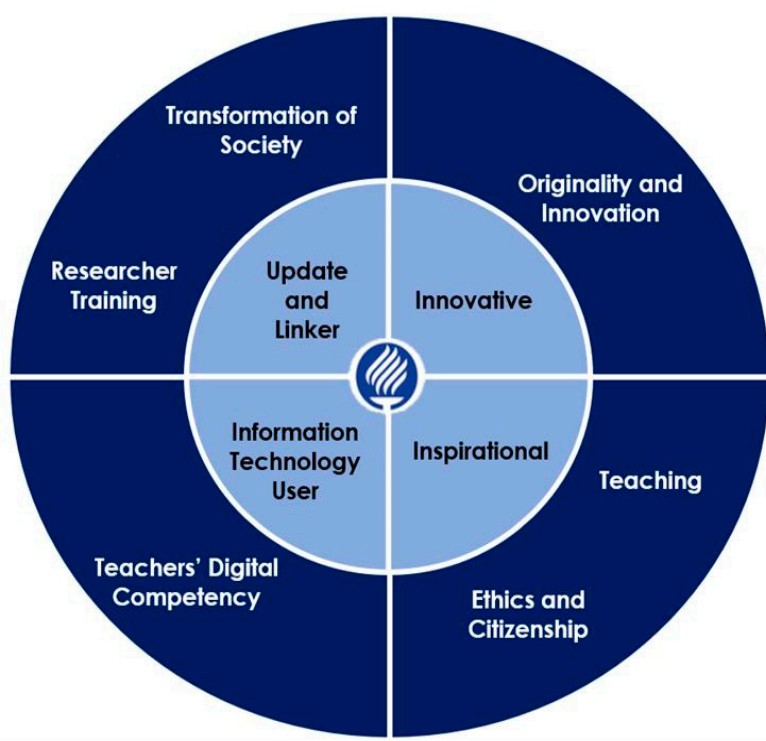

**Figure 1.** Model for Research Professors' Competencies. Source: Adapted from [20,21].

**Table 1.** Domains, Competencies, and Elements of Competencies for Evaluation of Research Professors.

| Domain | Competence | Elements of Competencies |
|---|---|---|
| A. Updated and linked. The professor is a recognized trainer in the discipline, constantly renewing and evolving, generating research that benefits society, integrating learning quickly into students' transformation process, through both professional activities and research. | A.1 Transformation of Society. It generates, applies and transfers knowledge for the solution of problems that transform society with a global focus and collaborates with a diversity of colleagues from an interdisciplinary approach, and brings in funding for research. | A.1.1 Participates as a member of a research faculty<br>A.1.2 Participates in interdisciplinary research groups<br>A.1.3 Publish scientific articles<br>A.1.4 Knows citations<br>A.1.5 Knows the H index<br>A.1.6 Makes congress presentations<br>A.1.7 Assesses papers at refereed conferences<br>A.1.8 Hosts refereed conferences or other academic meetings<br>A.1.9 Reviews articles/Edits peer-reviewed journals<br>A.1.10 Reviews scientific projects and programs<br>A.1.11 Obtains recognition<br>A.1.12 Attracts external funding for transdisciplinary research<br>A.1.13 Participates in leadership development<br>A.1.14 Has presence in the media |
| | A.2 Researcher training. The researcher-professors are able to develop researchers with specific research performances. | A.2.1 Thesis director<br>A.2.2 Students receive recognition<br>A.2.3 Students enter the National System of Researchers<br>A.2.4 Publishes with students in co-authorship<br>A.2.5 Students present at conferences<br>A.2.6 Develops other research professors |
| INN. Innovative. The professor generates high impact research, models, policies, products, and services in an accelerated manner for the benefit and transformation of society. | INN.1 Originality and Innovation. It makes exceptional discoveries that it manages to transfer and commercialize, it relates to ecosystems to promote the transfer of knowledge. | INN.1.1 Generates intellectual property: copyright and industrial property<br>INN.1.2 Generates applied knowledge<br>INN.1.3 Transfers knowledge |
| INS. Inspirational. The professor is a respected trainer who develops relationships with students and transmits to them passion for research and discovery, thus developing in them deep and meaningful learning. | INS.1 Teaching: positively influences the students' performance and encourages them to give results beyond course requirements; knows how to evaluate and innovate in education. | INS.1.1 Receives above-average teacher evaluations; students recommend him/her<br>INS.1.2 Receives above-average teacher evaluations; students describe him/her as encouraging<br>INS.1.3 Performs teaching activities<br>INS.1.4 Designs innovative strategies, courses, and programs |
| | INS.2 Ethics and citizenship: applies, promotes, and enforces codes of conduct and guidelines for ethics, sustainability, and social responsibility. | INS.2.1 Practices ethics and civic values for sustainable development<br>INS.2.2 Participates in community projects or associations with significant social impact<br>INS.2.3 Knows and applies the institution's guidelines |
| UTI. Information Technology User. The professor uses technology as an element that enables and empowers the transformation processes of people and the research they conduct. | UTI.1 Digital competence: transfers methodologies of use and implements processes of improvement and innovation according to the needs of the digital age of technology. | UTI.1.1 Participates and disseminates in scientific and professional networks<br>UTI.1.2 Uses technological tools to interact efficiently<br>UTI.1.3 Uses technological tools to support efficient learning |

Source: Own Source.

This study's purposes include:

(1) researcher-professors' self-evaluation of competencies that constitute objective, uniform, measurable, and aspirational parameters, according to international standards and supported by scientific information; (2) identification of these competencies' levels of performance and their relationships; and (3) knowledge of differences in competencies' performance between starting teachers and the National System of Researchers (in México, SNI) teachers and their relations with age. With their leader, professors with self-knowledge of their competencies can implement a development plan and, in the following year, implement strategies aligned with the institution's vision and their research objectives as a professor. After a year's implementation, they can measure their growth.

## 2. Materials and Methods

The study uses a mixed two-stage sequence design investigation: first, a qualitative design for the self-evaluation rubric; second, a non-experimental, quantitative design. The instrument's validity is exploratory.

### 2.1. Instrument

A preliminary list of criteria defined each competency with four levels of performance—initial, basic, autonomous, and consolidated—all written in first person and in positive terms. Then, a focus group using inter-judge methodology reviewed the 108 criteria conceptually and operationally. Those criteria were next submitted to three experts from each of the institution's six schools. Each expert belonged to the SNI and had more than 10 years' experience. They evaluated each competency's performance criteria according to content (uniqueness), relevance (items most closely related to the study object), and clarity (easily understandable, simple statements).

These expert evaluations' reliability was estimated according to [22]: reliability = total number of agreements/total number of coded units, with between-judge reliability considered acceptable at 0.85 and above. After elimination of unreliable items, a second focus group of six researchers from each school re-evaluated the remaining 79 criteria, using the same methodology. Matching and meaningful responses were incorporated to result in a final 33 criteria. We next administered a small pilot self-assessment to 50 professors. The resulting rubric describes all levels of performance, lowest to highest [23]. See into the "supplementary" file for a Spanish version of the rubric.

### 2.2. Procedure

This new self-evaluation tool was administered twice on an online employee portal from September 20 to December, 2018; and from 24 September to 2 January 2019, and professors responded based on available evidence. At the end of each self-diagnosis, an individual report was automatically sent to each professor.

### 2.3. Participants

Of 507 research professors, 320 completed the inaugural self-evaluation. Of these, 174 (55%) belonged to the SNI; 76 (31%) did not, but were becoming research professors; and 20% intended to become SNI members. Respondents (105 [33%] female; 215 [67%] male) had an average age of 44.50 years (DS 9.3), with a minimum of 28 and a maximum of 76. Of the professors, 67% held a doctorate, and 33% a master's degree. As for their disciplines, 38% belonged to the School of Engineering and Science, 16% to the School of Humanities and Education, and the remaining 46% to other schools. Table 2 displays exact frequencies and percentages.

### 2.4. Analysis

Exploratory factor analysis was used to compare the instrument's factor structure to application of the principal component extraction method [24]. To reveal relationships between competencies and their elements, using SPSS V 24 software, we analyzed respon-

dents' data to obtain descriptive statistics and correlations with Pearson's r coefficient. Table 3 qualifies the competence levels.

**Table 2.** Frequency and Percentage of Professors by School (N = 250).

| School | Frequency | Percent |
|---|---|---|
| ECSG | 18 | 5.6 |
| EHE | 50 | 15.6 |
| EIC | 120 | 37.5 |
| EM | 18 | 5.6 |
| EN | 35 | 10.9 |
| EAAD | 9 | 2.8 |

Source: Authors. Note: ECSG: School of Government; EHE: School of Humanities and Education; EIC: School of Engineering and Science; EM: School of Medicine and Health Sciences; EN: Business School; EAAD: School of Architecture, Art, and Design.

**Table 3.** Competence and Performance Levels of Research Professors.

| Competence Level | Performance Level |
|---|---|
| X < 1 | Initial |
| 1 < X < 2 | Basic |
| 2 < X < 3 | Autonomous |
| X > 3 | Consolidated |

Source: Authors.

## 3. Results

All scales attained acceptable levels [25], as shown in Table 4. To evaluate the instrument's reliability, we performed a Cronbach's alpha reliability analysis: values of Cronbach's alpha coefficients: $\alpha > 0.9$ is excellent, $\alpha > 0.8$ is good, $\alpha > 0.7$ is acceptable, $\alpha > 0.6$ is acceptable for scales with less than 10 items, and $\alpha > 0.5$ is poor [25].

**Table 4.** Cronbach's Alpha of Scale for Each Research-professor Competency.

| Competence | Cronbach's Alpha |
|---|---|
| Transformation of society (14 items) | 0.911 |
| Researcher training (6 items) | 0.871 |
| Originality and innovation (3 items) | 0.718 |
| Teaching (4 items) | 0.678 |
| Ethics and citizenship (3 items) | 0.523 |
| Teachers' digital competence (3 items) | 0.658 |

Source: Authors.

Barlett's sphericity test was applied to ensure that each competency's correlation matrix was meaningful ($p < 0.05$) and to be able to reject the hypothesis of independence of variables [26]. Kaiser-Meyer-Olin (KMO) sample adequacy measures were also obtained for each competency. To be acceptable, the KMO index must be greater than 0.5 [27], as shown in Table 5.

For the six competencies' KMO sample adequacy measures, feasibility of factor analysis was observed. For each competency, Bartlett's sphericity test was statistically significant ($p < 0.05$), resulting in rejection of the hypothesis of variables' independence.

In each competency's unifactorial structure, as observed in sedimentation graphs, the number of factors suggested by a self-value criterion greater than one (K1 rule) is one, and clearly, after the first factor, the slope stabilized. For each competency we obtained these criteria; in Figure 2, Transformations of Society; in Figure 3, Researcher Training; in Figure 4, Originality and Innovation; in Figure 5, Teaching; Figure 6, Ethics and Citizenship; and in Figure 7, Teachers' Digital Competence.

**Table 5.** KMO Sample Adequacy Measures and Bartlett's Sphericity Tests.

| Competence | KMO | Bartlett's Sphericity Test | | |
|---|---|---|---|---|
| | | $X^2$ | gl | *p*-Value |
| Transformation of society | 0.924 | 1473.422 | 91 | 0.001 |
| Researcher training | 0.869 | 572.419 | 15 | 0.001 |
| Originality and innovation | 0.659 | 150.023 | 3 | 0.001 |
| Teaching | 0.539 | 505.375 | 6 | 0.001 |
| Ethics and citizenship | 0.581 | 45.045 | 3 | 0.001 |
| Teachers' digital competence | 0.609 | 110.219 | 3 | 0.001 |

Source: Authors.

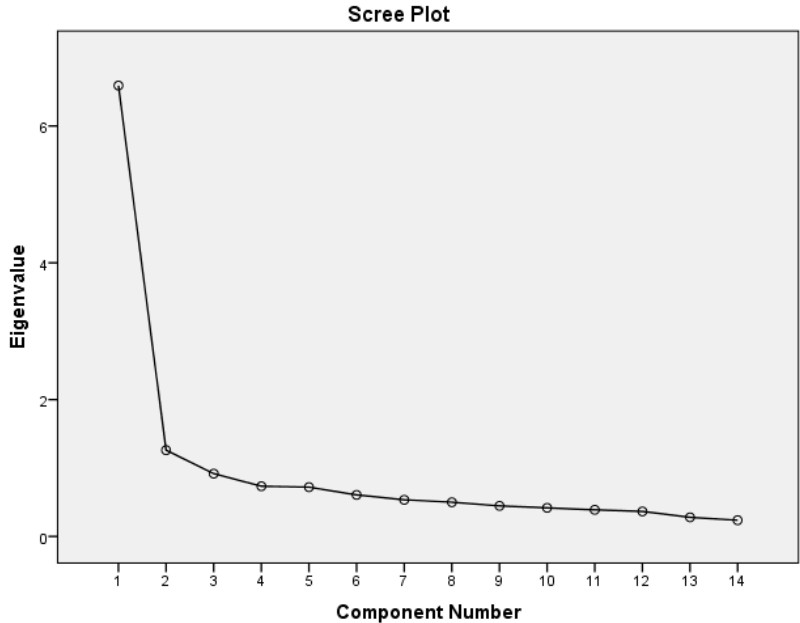

**Figure 2.** Sedimentation graph of the competence of Transformation of Society. Source: Authors.

**Figure 3.** Sedimentation graph of the competence of Researcher Training. Source: Authors.

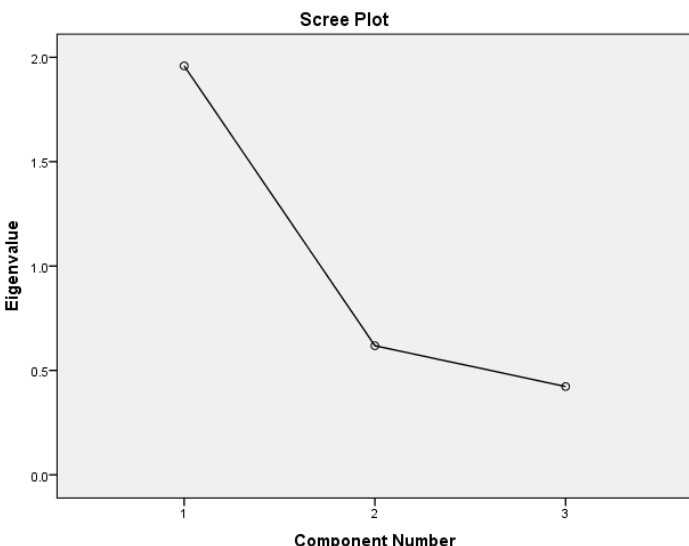

**Figure 4.** Sedimentation graph of the competence of Originality and Innovation. Source: Authors.

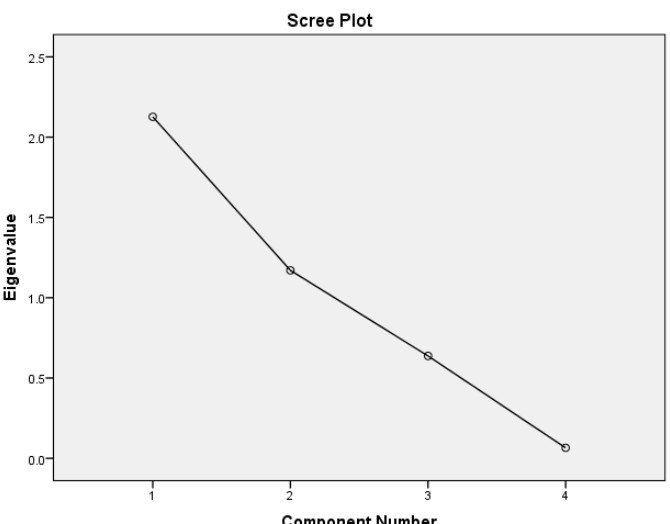

**Figure 5.** Sedimentation graph of the competence of Teaching. Source: Authors.

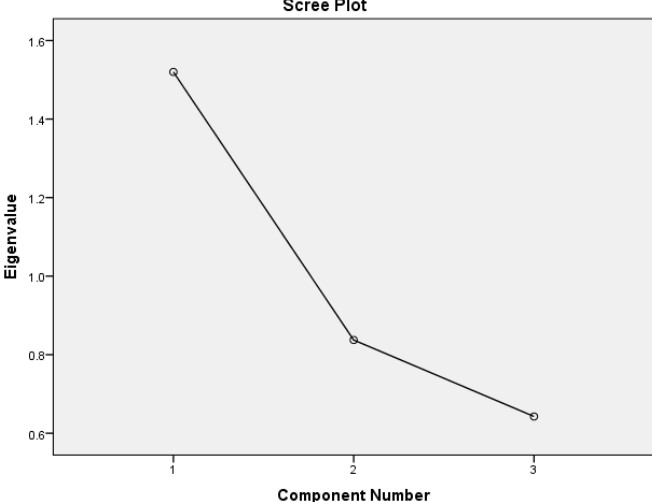

**Figure 6.** Sedimentation graph of the competence of Ethics and Citizenship. Source: Authors.

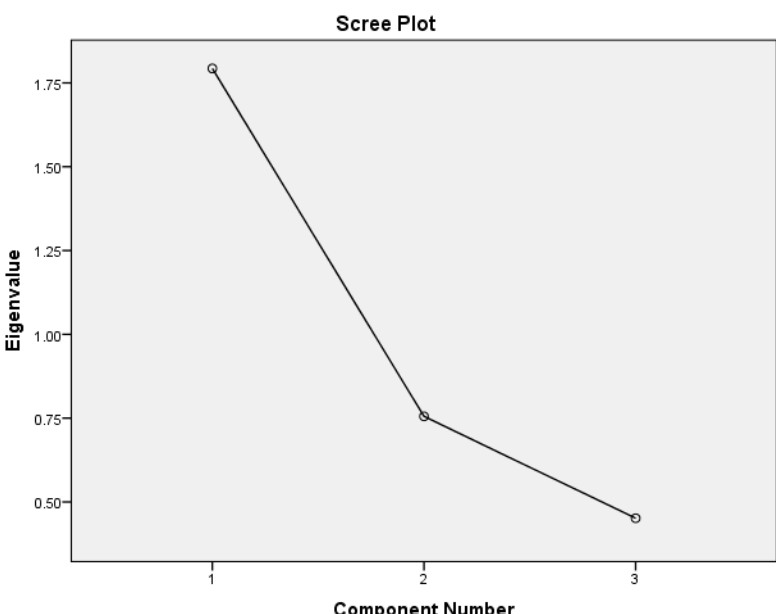

**Figure 7.** Sedimentation graph of the Teachers' Digital Competence. Source: Authors.

In Figure 5, the teaching competency's sedimentation graph does not fall off the slope, i.e., the competency is not explained by a single factor, perhaps because this competency is integrated with the professors' evaluation by students and by the professors' educational innovations. We suggest that in the future, these two competencies be separated. Table 6 displays percentages of variance explained by a first and second factor as well as factor loadings.

**Table 6.** Percentages of Variance Explained by First and Second Factors and Factor Loadings.

| Competence | Percentage of Variance Explained by a First Factor | Percentage of Variance Explained by a Second Factor |
|---|---|---|
| Transformation of society | 47.01% | 9.007% |
| Researcher training | 60.034% | 11.05% |
| Originality and innovation | 65.29% | 20.6% |
| Teaching | 53.1% | 29.26 |
| Ethics and citizenship | 50.66% | 27.91% |
| Teachers' digital competence | 59.76% | 25.18% |

Source: Authors.

The statistical test KMO = 0.906 indicates good adjustment of data to a factorial model [28], and Bartlett's sphericity test was statistically significant (X2 = 3807.925, $p$ = 0.000). Total variance was explained in six components, with the determinant = $1240 \times 10^{-8}$, as shown in Table 7.

*Main Results*

From the first evaluation in 2018 (Table 8), we can see the average of each competency. Research professors reported having an autonomous level of competence in teaching (X = 2.81; SD = 1.02). This competency is integrated by two factors, the professors' evaluations by students and by their teaching activity. In the institution, evaluation is very important; professors are recognized and provide recognition and economic incentives.

**Table 7.** Competencies' Performance Variance as Reported by Research Professors.

| | Total Variance Explained | | | | | | | | |
|---|---|---|---|---|---|---|---|---|---|
| Component | Initial Eigenvalues | | | Extraction Sums of Squared Loadings | | | Rotation Sums of Squared Loadings | | |
| | Total | % of Variance | Cumulative % | Total | % of Variance | Cumulative % | Total | % of Variance | Cumulative % |
| 1 | 11.060 | 33.516 | 33.516 | 11.060 | 33.516 | 33.516 | 7.766 | 23.534 | 23.534 |
| 2 | 2.471 | 7.487 | 41.003 | 2.471 | 7.487 | 41.003 | 3.258 | 9.871 | 33.406 |
| 3 | 1.781 | 5.398 | 46.401 | 1.781 | 5.398 | 46.401 | 2.515 | 7.622 | 41.028 |
| 4 | 1.584 | 4.799 | 51.200 | 1.584 | 4.799 | 51.200 | 2.293 | 6.948 | 47.976 |
| 5 | 1.452 | 4.400 | 55.600 | 1.452 | 4.400 | 55.600 | 2.048 | 6.206 | 54.182 |
| 6 | 1.244 | 3.768 | 59.368 | 1.244 | 3.768 | 59.368 | 1.711 | 5.186 | 59.368 |
| 7 | 0.999 | 3.029 | 62.397 | | | | | | |

Extraction Method: Principal Component Analysis. Source: Author

**Table 8.** Research Professors' Self-assessment of Competencies (N = 320).

| | Transformation of Society | Researcher Training | Originality and Innovation | Teaching | Ethics and Citizenship | Teachers' Digital Competence |
|---|---|---|---|---|---|---|
| Mean | 1.4738 | 1.1256 | 0.8374 | 2.8164 | 2.5925 | 2.2758 |
| Median | 1.3451 | 0.7986 | 0.5556 | 2.8125 | 2.6389 | 2.1667 |
| Std. Deviation | 1.02728 | 1.18743 | 1.03655 | 1.02709 | 1.17684 | 1.05261 |

Source: Authors.

Teachers' Digital Competence was reported at an autonomous level (X = 2.27; SD = 1.01) even though all professors receive training in using technological tools for instruction and for interaction with students. Professors' Originality and Innovation must be given a strong push because this competency was reported at the initial level (X = 0.837) as the lowest performing.

The bar graph (Figure 8) illustrates how Teaching rises above both Transformation of Society and Originality and Innovation for both research professors and each school.

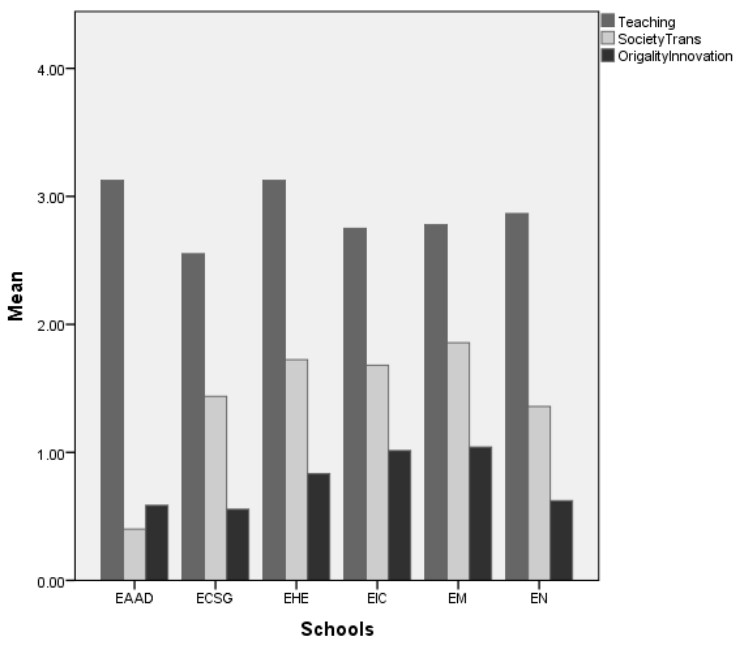

**Figure 8.** Teaching, Transformation of Society, and Originality and Innovation competency levels by school. Source: Authors.

Moreover, statistically significant differences in elements of competencies favor researchers except in Teaching and Ethics and Citizenship. In Table 9, we contrast the level of performance of research professors who are in the SNI and those who do not belong to the SNI. We recognize that performance in teaching and ethics for research and non-research teachers is similar because of empowerment provided to all professors. However, in the other competencies, teachers who belong to the SNI have a better level of performance, these differences being statistically significant.

**Table 9.** Researchers in the SNI: 0 = does not belong (N = 59); 1 = if applicable (N = 168).

| Competence | SNI | Mean | SD | t | *p*-Value | Elements | SNI | Mean | SD | t | *p*-Value |
|---|---|---|---|---|---|---|---|---|---|---|---|
| Updated and Linked | 0 | 0.831 | 0.753 | −5.910 | 0.000 | A.1 Transformation of society | 0 | 0.921 | 0.792 | −7.115 | 0.000 |
| | 1 | 1.72 | 1.081 | | | | 1 | 1.93 | 0.79 | | |
| | | | | | | A.2 Researcher training | 0 | 0.716 | 0.84 | −4.388 | 0.000 |
| | | | | | | | 1 | 1.51 | 1.08 | | |
| Innovative | 0 | 0.463 | 0.730 | −3.329 | 0.001 | INN.1 Originality and innovation | 0 | 0.542 | 0.926 | −2.852 | 0.000 |
| | 1 | 0.992 | 1.091 | | | | 1 | 0.993 | 1.084 | | |
| Inspiring | 0 | 2.837 | 0.851 | −0.381 | 0.704 | INS.1 Teaching. | 0 | 2.823 | 0.964 | −0.738 | 0.461 |
| | 1 | 2.784 | 0.930 | | | | 1 | 2.94 | 1.061 | | |
| | | | | | | INS.2 Ethics and citizenship | 0 | 2.851 | 1.077 | 1.272 | 0.205 |
| | | | | | | | 1 | 2.63 | 1.171 | | |
| Information Technology User | 0 | 2.063 | 0.889 | −2.977 | 0.003 | UTI.1 Teachers' digital competence | 0 | 2.022 | 0.89 | −3.329 | 0.000 |
| | 1 | 2.553 | 1.096 | | | | 1 | 2.54 | 1.096 | | |

Source: Authors. Note. Equal variances are assumed.

A significantly positive Pearson correlation coefficient greater than 0.4 was identified for Transformation of Society with different elements of other competencies. Publication of papers correlates with thesis assistance (r = 0.475, *p* < 0.000), with student admission to the SNI (r = 0.417, *p* < 0.000), with co-authorship (r = 0.628, *p* < 0.000), with students' speeches in congresses (r = 0.565, *p* < 0.000), and with dissemination of and participation in scientific networks (r = 0.423, *p* < 0.000). The research professors' age variable correlates with thesis advisor (r = 0.476, *p* < 0.000), acknowledgments to my students (r = 0.405, *p* > 0.000), and with student admission to the SNI (r = 0.469, *p* < 0.000).

One Year Post Implementation

Of the 30 research professors (63% male; 37% female) who completed the follow-up self-assessment, all had doctoral training, 54% were SNI professors, and 41% were from the EIC, 23% the EHE, 17% the BE, 10% the EM, and the rest from other schools. All had elaborated their objectives with their leaders and worked to implement their development plans. This approach reflected growth in all competencies, and although the growth was not significant, no competencies decreased. Visually we can see the previous and current performance in the spider web graph in Figure 9.

After one year of working towards their objectives, professor-participants attained consolidated performance in both the Teaching and Ethics and Citizenship competencies and autonomous performance in Teachers' Digital Competence. In contrast, the great challenge is Originality and Innovation, which remains below the initial level. We must also work on increasing Researcher Training and Transformation of Society, which remained at the initial level. We need to intervene so that teachers focus their goals towards the competencies they have in initial levels in a more vigorous way.

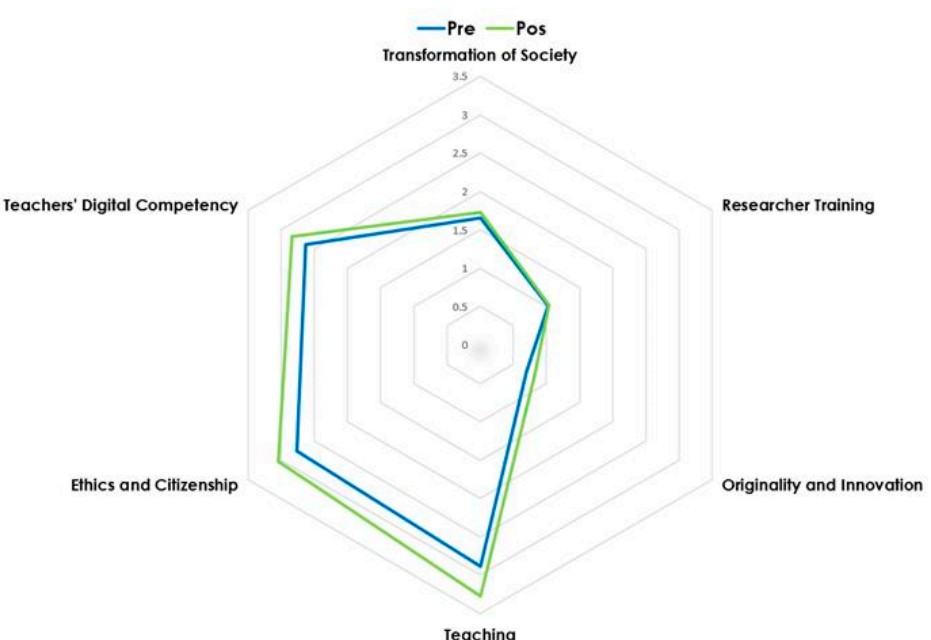

**Figure 9.** Comparison of professors' performance by competency after one year.

## 4. Discussion and Conclusions

This study emerged from the need for in-depth identification and evaluation of high-performing research professors' competencies with the intention of developing internationally recognized professors. We recognize that, as other research has done [4], that techniques to measure the development of 'soft' skills and competencies were not well developed. Additionally, it is widely recognized that this is a complex area with no simple indicators.

In the evaluation instrument's inaugural year, 320 research professors completed the self-evaluation, we found it to have construct validity. However, from our findings, we recommend separating the professors' evaluation from their academic work. Even so, the evaluation model is valid and reliable for all four dimensions: updated and linked, innovative, inspirational, and user of information technology. Not surprisingly, productivity and funding performance are linked, so we see how these two competencies' factorial loads develop together rather than independently. Thus, we have a good self-assessment with appropriate psychometric properties, which can be used in similar contexts for further research. Even so, competency implies deployment of complex performances, procedures, and attitudes to address a particular situation.

From the 30 professors' second self-assessment a year later, we have the following findings.

### 4.1. Transformation of Society

This competency shows the need for greater support, empowerment, and resources for research professors' productivity ($X1 = 1.65$; $X2 = 1.72$). As one of the most important variables in performance evaluation [29], this competency remains at the basic level. In this model, the competency is determined by the following elements: affiliation with a research group; number of articles as first author, combined with journal quality; number of citations, H index, presentations at high-quality conferences, reviewing papers for peer-reviewed conferences, organizing peer-reviewed conferences or other academic meetings, reviewing articles; leadership in the discipline; recognition; and attraction of research funds. In this context, each school and each professor must define an estimate to increase productivity, specifically, the number of citations, revisions of indexed and peer-reviewed articles, and attraction of funds. Results can foster decisions about a strategy for vision and development.

### 4.2. Training of Researchers

Professors declared the need for greater support of researchers-in-training (X1 = 1.02; X2 = 1.042). Research professors are aware that they should challenge their student trainees in the following: participation in academic debate; preparation of articles for publication; development of attitudes toward research experience; guide trainees to publish more, attend more congresses, win research awards, and obtain admission to the SNI. Effective research training tends to involve collaborations with a degree of reciprocity because both parties receive benefits. We are thinking of implementing a mentoring training for the training of researchers with the support of a platform.

### 4.3. Originality and Innovation

In both reporting periods, unfortunately, this competency did not reach the initial level (X1 = 0.69; X2 = 0.81). Research professors need the following: training and empowerment in generating intellectual property; generating applied knowledge; transferring that knowledge for commercialization; and institutional support for links with companies and organizations in entrepreneurial projects that potentially generate knowledge transfer and commercialization. We are also generating specific training in technological entrepreneurship for teachers and researchers in training with the support of entrepreneurship models linked to business and industry.

Similar studies mention institutions that are not adapting quickly enough to the needs of industry or the expectations of potential students [4].

### 4.4. Teaching

From the first to the second year, this competency rose from the intermediate to the strategic level (X1 = 2.89; X2 = 3.28). Evaluating and improving students' didactics, satisfaction, and intellectual attainment are crucial to the institution. In cases of poor student evaluations, a support program is implemented for the instructor; if poor evaluations continue, instructors can be dismissed. In educational innovation, professors reported an autonomous level of competence despite being constantly enabled with various strategies and didactics to implement innovative course design and program development.

### 4.5. Ethics and Citizenship

Here professors reported an intermediate level of application and promotion of ethical codes and citizenship values as well as institutional conduct guidelines (X1 = 2.75; X2 = 3.04). The institution provides courses and practices for knowledge of institutional guidelines and codes of ethics, along with initiatives for participation in volunteer community service projects. More emphasis is needed to help raise the competency of instructors at the basic level.

### 4.6. Teachers' Digital Competence

In general, professors report themselves at the autonomous level, with intensive daily use of technology in classroom learning support and access to state-of-the-art infrastructure for instructional practice (X1 = 2.62; X2 = 2.83).

For those who perceive themselves at a basic level, working more on research dissemination and participation in scientific and professional networks is necessary.

Because consolidated professors have developed most competencies in a very intensive way, they ranked higher than initial research professors in all competencies except Teaching and Ethics and Citizenship. Transformation of Society correlates with various elements of the same competency and with research-professor development in significantly positive Pearson's coefficients greater than 0.4, but it does not correlate with other competencies' elements. Age correlates significantly (greater than 0.4) with thesis advising, recognition of their students, and with their students joining SNI.

Overall, from professor-respondents' comments, we observed that it was very good for researchers to receive immediate responses with feedback and recommendations on

how to improve competencies and skills and what the next level of performance should be—in addition to the knowledge and information held per professor, per school, and per institution.

We consider for future research that it is necessary for more professors to be engaged in answering the self-diagnosis in the two periods during the year, before and after. If they do not answer any of these two periods, then we cannot make any comparisons of performance. In the calculated periods, we missed many professors, as some only answered at the beginning and others only answered at the end of the period. This year, 2021, the diagnosis was considered as an element of its teacher-training program, and we believe that we will have more participants.

It is also necessary to state that each professor should work with his or her leader in developing objectives corresponding to the competencies where there are weaknesses, and without forgetting the strong areas. We know that this work is a work of the professor's introspection and is particular of the disciplines and research area. As other research has found [4,8,17], we know that personal decision making is very important as researchers are inherently a very diverse group of individuals, with diversity arising not only from their highly specialized topics of study but also their diverse modes of operation and personal needs and backgrounds, but it must be approached with a holistic and institutional vision. We work in order to improve the understanding of the importance of more formalized training and career development for all researchers.

For future research, we integrate questions to the self-diagnosis about the impact that the pandemic due to COVID-19 will have on their research, projects and funding. Surely, this situation will also have consequences on the development of professors' competencies, which are not exactly negative. It will be very interesting to know their implications.

**Supplementary Materials:** The following are available online at https://www.mdpi.com/1999-5903/13/2/41/s1.

**Author Contributions:** Investigation, G.T.D. and N.H.-G. All authors have read and agreed to the published version of the manuscript."

**Funding:** This research received no external funding.

**Data Availability Statement:** Not Applicable, the study does not report any data.

**Acknowledgments:** The authors would like to acknowledge the financial support of Writing Lab, Institute for the Future of Education, Tecnologico de Monterrey, Mexico, in the production of this work.

**Conflicts of Interest:** The authors declare no conflict of interest.

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
