# Peer review of "Research Professors’ Self-Assessment of Competencies"

_futureinternet, doi:10.3390/fi13020041_

Round 1
Reviewer 1 Report
The article should be improved in the following respects:
1. It should improve the inclusive pedagogical approach in the theoretical-conceptual framework.
2. Methodological design is appropriate, but it is necessary to increase the level of reflection and criticism, i.e. not only to describe, but to make an effort of thoughtful interpretation.
3. The conclusions should improve substantially, as well as the discussion itself, which is that the contrast and comparison of the results found with other similar studies and research papers should be increased.
Finally, the updating and degree of internationalization of bibliographic references must be greatly improved.
Author Response
Thank you very much for your comments, we have reviewed each point:
1. It should improve the inclusive pedagogical approach in the theoretical-conceptual framework. Elements of the task of the research teacher are described.
2. Methodological design is appropriate, but it is necessary to increase the level of reflection and criticism, i.e. not only to describe, but to make an effort of thoughtful interpretation. We integrate reflections in the conclusions and introduction.
3. The conclusions should improve substantially, as well as the discussion itself, which is that the contrast and comparison of the results found with other similar studies and research papers should be increased. We add some comparisons.
4. Finally, the updating and degree of internationalization of bibliographic references must be greatly improved. International references have been significantly increased and updated.
Thank you for your comments.

Reviewer 2 Report
Once the authors have made these changes, I think the article is publishable.
A table or figure should never appear without reference in the text prior to it (you have many figures and tables without referencing), never in the text after it appears.
In the first two paragraphs of point '1.1 Framework', a series of data is cited, but the source does not have its corresponding reference in the 'References' section. You must put the number of the citation in brackets in the two paragraphs, and the reference in the 'References' section.
The fourth section (Line 245) should be called ‘Discussion and Conclusions’.
And in that same fourth section, you should add a paragraph at the end that talks about the limitations and includes deductions for future research.
Author Response
Thank you for all your comments, we review every point:
1. Once the authors have made these changes, I think the article is publishable. Thanks for your consideration
2. A table or figure should never appear without reference in the text prior to it (you have many figures and tables without referencing), never in the text after it appears. We change these and refer to each table and figure beforehand.
3. In the first two paragraphs of point '1.1 Framework', a series of data is cited, but the source does not have its corresponding reference in the 'References' section. You must put the number of the citation in brackets in the two paragraphs, and the reference in the 'References' section. We place the corresponding reference.
4. The fourth section (Line 245) should be called ‘Discussion and Conclusions’. Done
5. And in that same fourth section, you should add a paragraph at the end that talks about the limitations and includes deductions for future research. We add two paragraph with limitations and another with implications for future research
We add the document with its considerations
Thank you

Reviewer 3 Report
This is a very welcome research paper looking at a field not easy to analyse and discuss mainly because of the general reluctance of faculty for introspection. However, the genuine interest of the researchers and their clear long-time interest in the field make the result highly relevant. And the areas of the researchers’ interests have been finely selected in spite of their relative difficulty to document: “leadership in a discipline, production of quality research, innovation, teaching, ethics, citizenship, use of technology, and linkage, funding, and training of other researchers—all focused on solving society’s current and future problems.” (lines 34 – 37).
There is one aspect that I feel might improve the understanding of the background of the research: the clarifying of the use of the term “professor”. Is it a general umbrella term for faculty or does it refer mainly to the specific, top, tenured, higher education position relative to other faculty titles? Plus, occasionally the term “teacher” shows also up (as in Table 1, line 83)
There are also some minor issues that need to be improved:
Line 25 – recently refers to a paper from 2008. Isn’t there anything newer?
The link in the reference 4 (lines 346 – 347) is no longer valid. Similar references may be however found - https://repository.lboro.ac.uk/articles/report/Review_of_progress_in_implementing_the_recommendations_of_Sir_Gareth_Roberts_regarding_employability_and_career_development_of_PhD_students_and_research_staff/9354350/1
In Table 1 (line 83) – last paragraph there are some typos:
empowers the transfor0 mation processes (column 1)
UTI.1 Teacher´ digital competence (column 2)

Author Response
Thank you very much for your comments, we have reviewed each point:
1. This is a very welcome research paper looking at a field not easy to analyse and discuss mainly because of the general reluctance of faculty for introspection. However, the genuine interest of the researchers and their clear long-time interest in the field make the result highly relevant. And the areas of the researchers’ interests have been finely selected in spite of their relative difficulty to document: “leadership in a discipline, production of quality research, innovation, teaching, ethics, citizenship, use of technology, and linkage, funding, and training of other researchers—all focused on solving society’s current and future problems.” (lines 34 – 37). Thanks for your comments
2. There is one aspect that I feel might improve the understanding of the background of the research: the clarifying of the use of the term “professor”. Is it a general umbrella term for faculty or does it refer mainly to the specific, top, tenured, higher education position relative to other faculty titles? Plus, occasionally the term “teacher” shows also up (as in Table 1, line 83). Thanks for your comments and we clarify and define the concept of the research professor.
3. There are also some minor issues that need to be improved:
Line 25 – recently refers to a paper from 2008. Isn’t there anything newer? Yes we include new works about
The link in the reference 4 (lines 346 – 347) is no longer valid. Similar references may be however found - https://repository.lboro.ac.uk/articles/report/Review_of_progress_in_implementing_the_recommendations_of_Sir_Gareth_Roberts_regarding_employability_and_career_development_of_PhD_students_and_research_staff/9354350/1 we include this, excellent work
In Table 1 (line 83) – last paragraph there are some typos:
empowers the transformation processes (column 1) thank you, we correct typos
UTI.1 Teacher´ digital competence (column 2) thank you, we correct typos
We add the document with its considerations
thank you
